# Patients' preferences in dental care: A discrete-choice experiment and an analysis of willingness-to-pay

Susanne Felgner 📧 *, Cornelia Henschke

Department of Health Care Management, Berlin Centre of Health Economics Research (BerlinHECOR), Technische Universität Berlin, Berlin, Germany

* susanne.felgner@tu-berlin.de

## Abstract

### Introduction

Dental diseases are a major problem worldwide. Costs are a burden on healthcare systems and patients. Missed treatments can have health and financial consequences. Compared to other health services, dental treatments are only covered in parts by statutory health insurance (SHI). Using the example of dental crowns for a cost-intensive treatment, our study aims to investigate whether (1) certain treatment attributes determine patients' treatment choice, and (2) out-of-pocket payments represent a barrier to access dental care.

### Methods

We conducted a discrete-choice-experiment by mailing questionnaires to 10,752 people in Germany. In presented scenarios the participants could choose between treatment options (A, B, or none) composed of treatment attribute levels (e.g., color of teeth) for posterior (PT) and anterior teeth (AT). Considering interaction effects, we used a D-efficient fractional factorial design. Choice analysis was performed using different models. Furthermore, we analyzed willingness-to-pay (WTP), preference of choosing no and SHI standard care treatment, and influence of socioeconomic characteristics on individual WTP.

### Results

Out of n = 762 returned questionnaires (response rate of r = 7.1), n = 380 were included in the analysis. Most of the participants are in age group "50 to 59 years" (n = 103, 27.1%) and female (n = 249, 65.5%). The participants' benefit allocations varied across treatment attributes. Aesthetics and durability of dental crowns play most important roles in decision-making. WTP regarding natural color teeth is higher than standard SHI out-of-pocket payment. Estimations for AT dominate. For both tooth areas, "no treatment" was a frequent choice (PT: 25.7%, AT: 37.2%). Especially for AT, treatment beyond SHI standard care was often chosen (49.8%, PT: 31.3%). Age, gender, and incentive measures (bonus booklet) influenced WTP per participant.

**Data Availability Statement:** An anonymized minimal data set of the choice analysis is included as Supporting Information. This data set complies with the requirements of the ethics committee and the institution's data protection officer's

requirements. Individual participant data may only be presented in aggregated form according to the ethics application. The ethics application has been accepted by the ethics committee of the Charité Universitätsmedizin Berlin (application no. EA4/109/19; contact details: Charité – Universitätsmedizin Berlin, Ethikkommission der Charité, Charitéplatz 1, 10117 Berlin).

**Funding:** This study was funded through the Berlin Centre for Health Economics Research by the German Federal Ministry of Education and Research (grant no. 01EH1604A), URL: www.bmbf.de. Both authors are grant recipients. The funder had no role in study design, data collection and analysis, decision to publish, or preparation of the manuscript.

**Competing interests:** The authors have declared that no competing interests exist.

## Conclusion

This study provides important insights into patient preferences for dental crown treatment in Germany. For our participants, aesthetic for AT and PT as well as out-of-pocket payments for PT play an important role in decision-making. Overall, they are willing to pay more than the current out-of-pockt payments for what they consider to be better crown treatments. Findings may be valuable for policy makers in developing measures that better match patient preferences.

## Introduction

Understanding how patients assess various aspects of health care interventions is important for clinical, coverage, and policy decisions. As a result, considering patient preferences in health care decision-making and policy can improve utilization of interventions and public health programs, satisfaction with those, and patient adherence to finally improve effectiveness of health care services [1]. To identify preferences for various attributes of an intervention, stated preference methods such as the discrete-choice approach can be used, particularly to quantify stakeholders' preferences in health care. At the same time, this approach offers a mechanism for patients to participate in decision-making and may facilitate shared decision-making. In practice, the discrete-choice approach is also used to estimate willingness-to-pay for attributes, which is especially beneficial in the case of treatments where high co-payments may arise such as oral health services. Preferences of utilization of oral health services from patients' perspective have rarely been studied so far, although chronic and untreated dental diseases (e.g., caries) can lead to serious consequences such as pain, sepsis, reduced quality of life, and work productivity. This places a burden on patients in various aspects of their lives and on healthcare system in terms of capacity [2].

Although in Germany a broad range of oral health services is covered, high out-of-pocket payments may occur for patients. Regarding dentures, for instance, statutory health insurance (SHI) covers 60% of the standard care costs, which can be defined as broad coverage compared to other countries [3, 4]. Incentive measures, such as regular dental check-ups within the last five or ten years before respective treatment, may increase the SHI's coverage [5]. Additionally, patients may take out private dental supplementary insurances to reduce out-of-pocket payments [6], and choose treatments beyond defined standard care. Nevertheless, perceived unmet needs in dental care still exist. Households with highest budget (fifth household budget quintile) take higher out-of-pocket payments than households with lowest budget (first household budget quintile) [7].

While costs of dental treatments seem to play a major role in patients' decision-making [8], the choice of a treatment may also be influenced by individual preferences regarding further attributes. Patients may value certain treatment attributes differently [9, 10], such as color of a dental crown. Former studies have investigated patient preferences for dentures [11], caries prevention measures [12], and willingness-to-pay for medical tourism [13]. We focus on a prosthodontic treatment–the placement of a full dental crown–due to high variability in options and costs to be borne by the patients themselves. In Germany, SHI covers a fixed subsidy of 50% (60% as of 10/2020) for standard treatment of dental crowns. The remaining 50% (40%) have to be paid out of pocket by patients, plus the difference of costs when choosing superior materials. The attributes out-of-pocket payment and aesthetics vary greatly between

different dental crown treatments. These assumed (un)desirable attributes for patients move proportionally against each other, i.e., an aesthetically pleasing dental crown is expensive and vice versa. The SHI alternative may implicate an aesthetically unattractive result to patients (i.e., darker-colored not natural appealing dental crown). To our knowledge, this treatment has not yet been studied for the German health care system using an experimental approach. This study investigates patient preferences presented by benefit allocations to treatment attributes for dental crown treatments in Germany addressing the following research questions:

1. To what extent do patients assign benefits to attributes of dental crown treatments and how does this influence their choice behavior?

2. Can out-of-pocket payments be considered a barrier to patients' access when deciding for a dental treatment?

## Methods

Discrete-choice experiments (DCE) are an established instrument particularly in health sciences [14] for measuring patient preferences in their choice behavior by estimating benefit assignments, and for calculating willingness-to-pay (WTP) taking out-of-pocket payments into account. Although, medical professionals usually recommend a treatment option, due to restricted coverage in oral health care services, final decisions are largely guided by patient preferences. A DCE is best suited to collect data for analyzing preferences of patients and their WTP. This is especially the case as patients have to decide between different treatment opportunities that come along with large variations in out-of-pocket payments. To analyze patient preferences, we therefore conducted a DCE. Furthermore, we analyzed overall (and individual) WTP to compare monetary value of respondents' willingness-to-pay and the SHI out-of-pocket payment for each attribute. In addition, we conducted regression analyses to calculate the relationship between socio-economic and other characteristics of participants and defined decision variables. Descriptive analyses were used to illustrate quantitative results (S3, S4 and S6 Files).

### Experimental design & questionnaire

Prior to the study, a systematic literature review [15] and focus group interviews [8] were conducted to identify attributes that influence patients in their choice for or against dental treatments. In the DCE dental treatments are presented as a combination of attribute levels in choice sets. As those attributes and levels should be plausible, and clinically relevant, being as realistic as possible [16–18], we used most relevant treatment attributes identified. Levels for aesthetics, compatibility, durability [19], and out-of-pocket payment [5] were determined by research. We differentiate between two teeth areas "posterior teeth" (PT) and "anterior teeth" (AT), since different patient preferences can be assumed for it [20]. Attributes and levels were presented to (potential) participants in the questionnaire (S1 File: Questionnaire). For the attribute aesthetics, an extra document was created for visualization (S2 File: Document "Aesthetics"). **Table 1** gives an overview of the treatment attributes and its levels.

Considering four attributes (x1-x4) with 3, 2, 4, and 4 levels, n = 9,216 possible choice sets resulted in a full factorial design (3x2x4x4 = 96; 96x96). Since this cannot be answered by an individual participant, a fractional factorial design was used [21] (S1 Table: Design output). Aiming at a 100% D-efficient design [21], and assuming interaction effects between the attributes x1 and x4, and x3 and x4, n = 96 choice sets were necessary (S2 Table: Calculation of D-efficiency). Ensuring that the questionnaire was manageable for our study participants [1], n = 12 questionnaire blocks were formed and randomly assigned to the participants [17]. For

**Table 1. Treatment attributes and its levels.**

| Attribute | Definition | Levels | |
|---|---|---|---|
| 1. Aesthetics | In terms of appearance, result of treatment individually perceived as beautiful. This attribute describes the visibility of a dental crown. | ✓ Natural color<br>✓ Lightly visible<br>✓ Strongly visible | |
| 2. Compatibility | Intolerance reaction of human body due to dental material in form of an allergic or a local toxic reaction[1]. | ✓ No risk<br>✓ 1 out of 10,000 people with allergic or local toxic reaction | |
| 3. Durability | Expected length of time from completion of a treatment to another new treatment that is medically or technically necessary. | ✓ 5 years<br>✓ 10 years<br>✓ 15 years<br>✓ 25 years | |
| 4. Out-of-pocket payment | Costs that must be paid by patient for dental crown treatment. The co-payment taken by health insurance has already been subtracted here. | **Posterior teeth**<br>✓ 50 €<br>✓ 150 €<br>✓ 450 €<br>✓ 600 € | **Anterior teeth**<br>✓ 50 €<br>✓ 200 €<br>✓ 450 €<br>✓ 600 € |

[1] In rare cases (1 in 10,000 people), an intolerance reaction, i.e., an allergic or local toxic reaction, may occur. Allergies are characterized by symptoms such as dry mouth, toothache, and receding gums to discomfort in the throat, lip eczema or rash on the face. Local toxic reactions are non-allergic inflammations of the oral mucosa in the immediate vicinity of the tooth crown. In the cases described, depending on the severity of the clinical symptoms, the "problematic material" must be replaced or (the dental crown) removed completely.

the design calculation we used the statistical software SAS (version 9.2). The *%ChoicEff* macro was used to create the experimental design, considering interactions as constraints via *restrictions = option*, and the *%MktEx* macro was used to create the choice sets. Blocked design was realized via the *%mktblock* macro [22].

The paper-based questionnaire was divided into an introductory part including the informed consent and explaining information on attributes and levels (part A), the choice scenarios (choice sets) to be answered by the participants (part B), and questions regarding the participants' (socioeconomic) characteristics (part C). Alternatives of dental crown treatment were presented in eight choice sets each, separated in part B1 focusing on PT and B2 on AT. Studies report that experiments including up to n = 32 scenarios are manageable by participants [1, 23]. This is in line with our study using n = 18 choice scenarios per questionnaire: n = 8 choice sets each for PT and AT, plus two additional sets to test reliability ("double question") and validity ("clear question") of responses (T = (2x8)+2 = 18). For the "clear question", the two-alternative-choice sets included only the best vs. worst attribute levels (e.g., 25 vs. 5 years durability). It can be assumed that only participants who understood the questionnaire's content correctly answered this question appropriately. The participants were asked to select their preferred alternative with its attributes and levels.

In reality, and particularly in health care, individuals face non-binary multiple choices [24]. For this reason, we created choice sets consisting of two unlabeled alternatives (A & B), and the option of "no treatment (opt-out)". An unlabeled design allowed us to assign attributes to the alternatives without being oriented to a defined treatment [25], and intended to reduce a possible bias. Including opt-out was necessary to create real life scenarios, and to explore patients' reasons against a treatment [25]. An example of a choice set can be seen in **Table 2**. Collected participants' (socioeconomic) characteristics are, e.g., age, income, insurance and oral health status. Furthermore, importance of the four treatment attributes was assessed for PT and AT via a 5-point Likert scale (dimension: very important–not important).

## Data collection

The study was conducted in the German federal states Berlin and Brandenburg. Since it is more likely for patients at a more mature age to have experiences with dental crown treatments and to have significant financial resources of their own [13, 26], we addressed people aged 30 and older. We aimed at an equal distribution of urban and rural population. By prevailing conditions, number of districts and counties in these areas are similar [27]. Household incomes in Berlin and Brandenburg are the same overall and are approximately equally distributed among Brandenburg's counties [28], with incomes in both states close to the national average [29]. Furthermore, we aimed at women and men equally distributed. In compliance with the European General Data Protection Regulation, address data of potential participants, considering the minimum age and an equal gender distribution, were requested according to §34 of the Federal Registration Act from residents' registration offices of the city of Berlin and selected counties in Brandenburg. For organizational reasons, the study's catchment area was limited to these states. For the state of Brandenburg, one registration office per district was randomly selected. The number of contacted registration offices was thus limited to at least 19 institutions. Potential participants, either from urban or rural areas, were randomly assigned to one of the twelve questionnaire blocks using the Software RStudio.

We used the rule of thumb by Orme [30] ((n x t x a)/c > = "500 to 1,000") to determine the sample size n for the DCE, creating a (minimum) recommended level of participants. For calculating the number of choice sets (t), the number of treatment alternatives per choice set (a), and the highest number of levels (c) were considered: t(A) = t(B) = 8, a = 2, and c = 3x4 = 12. The calculated sample size was n = 750 (minimum n = 375). Assuming a response rate of questionnaires of r = 7% resulting from further studies experiences at our department, the calculated number of questionnaires was n = 10,715. Rounding up the results and considering an equal distribution of questionnaires among urban and rural areas, a total of n = 10,752 questionnaires were sent out by mail.

Before the survey start, a pretest was conducted with n = 15 participants of diverse educational backgrounds and ages. Based on this, a few minor linguistic corrections for the questionnaire's comprehensibility followed and a processing time of about 20 minutes was set. The final survey included the following documents: (1) questionnaire and cover letter, declaration of participation, and participant information, (2) extra document "Aesthetics", and (3) free-return envelope addressed to the department. As an incentive for participating in pretest and survey, 20 shopping vouchers of 50 € each, were raffled in a lottery. **This study was approved by the ethics committee of the Charité Universitätsmedizin Berlin (application no. EA4/ 109/19).**

**Table 2. Example of choice set.**

| 1. Choice "anterior teeth" | | |
|---|---|---|
| **Attributes** | **Treatment A** | **Treatment B** |
| 1. Aesthetics | strongly visible | natural color |
| 2. Compatibility | 1 out of 10,000 people with allergic or local toxic reaction | no risk |
| 3. Durability | 10 years | 25 years |
| 4. Out-of-pocket payment | 50 € | 200 € |
| **I choose . . .** | . . . treatment A. ☐ | . . . treatment B. ☐ |
| | . . . none of the treatments. ☐ | |

## Data editing & coding

According to predefined criteria, completed questionnaires were included for further consideration if (i) the declaration of participation was confirmed. Questionnaires were excluded, when (ii) failing the plausibility check: (a) "double question" was not answered in the same way, and (b) "clear question" was answered irrationally. Furthermore, questionnaires were excluded if (iii) >50% of the choice sets were not answered in section B1 and B2, (iv) the participants did not answer the question on their age or with "under 30 years", (v) the question on insurance coverage was not answered, or with "I don't know", or "private health insurance", and (vi) the question on gender was not answered.

Regarding choice analysis the data set was effect coded which is recommended when an opt-out alternative is used [31]. Negative levels were selected as reference levels (and positive for WTP analysis) assuming to be unattractive for patients, i.e., strongly visible, risk of incompatibility, shortest duration (5 years), and highest costs (600 €). The reference level was coded with a value of -1. For all attributes of the opt-out, the very low value "-9999" was set because sum of benefit values would result in zero [32]. For calculation of the WTP over all participants the cost attribute was not effect coded but had continuous coding for more interpretable values [33]. For further analyses single variables were dummy coded, e.g., gender.

Since individual WTP cannot be estimated within a DCE [34], we defined a variable WTPmax_PT and _AT presenting the highest level value of the attribute out-of-pocket payment for a chosen treatment alternative across all alternatives per participant and teeth area. If opt-out was selected, we considered 0 €. To examine patients' behavior with respect to their choice between (I) "treatment" and opt-out, and (II) "SHI standard care" and "treatment beyond SHI standard care (SHI+)", we created further dependent variables as part of the choice analysis and depicted frequencies per participant and teeth area. The variable for "treatment" comprised the frequency of chosing treatment A or B in the choice set and "no treatment" comprised choice of opt-out. In the variable "SHI+" only choice sets with levels >SHI standard care of the attributes aesthetics and out-of-pocket payment were included: (a) lightly visible or natural color, and 450 € or 600 € for PT, and (b) natural color, and 450 € or 600 € for AT. Combinations of these levels are given in some choice sets of each questionnaire block.

## Choice analysis & willingness-to-pay analysis

In Lancaster's [35] and McFadden's [36] random utility theory, it is assumed that the actual utility of a choice set is not directly observable. The total utility of a set is composed of observable and non-observable components. It is assumed that an individual chooses the alternative with a combination of attributes from which she or he has the greatest utility over the other selectable alternatives. An indirect utility function was estimated that represents the expected observable utility (V) for a person, and is composed of a combination of (non-)observable random components as error term (ε) [37–39]. We specified the following utility function, in which the participants' preferences for the attributes are captured and different utility allocations among attributes can be examined as a function of the participants' (socioeconomic) characteristics. The utility function V for individual i and alternative j in choice set s is to be expressed, in terms of the attributes of the alternatives (X) and characteristics of the participants (Z), as:

$$U_{ijs} = V_{ijs} + \varepsilon_{ijs} = X_{ijs}{}'\beta_j + Z_i{}'\gamma + \varepsilon_{ijs}$$

We assume the non-observable component is parametrically distributed and thus use a probalistic analysis of individual choice behavior [38]. The probability of choosing between given

alternatives (J) is as follows:

$$P_{ij} = Prob\Big( U_{ji} > U_{Ji} \ \forall \ J \neq j \Big) \ = Prob(V_{ji} + \varepsilon_{ji} > V_{Ji} + \varepsilon_{Ji} \ \forall \ J \neq j)$$

Assuming that the error term is extreme-distributed, the probability of choosing alternative j, presenting the standard logit specification [38], is:

$$L_{ij} = e^{\text{\ss} x_{ij}} \Big/ \sum_J e^{\text{\ss} x_{iJ}}$$

letting

$$V_{ji} = \text{\ss} x_{ij}$$

The model of utility for an individual i choosing a treatment alternative j can be estimated as:

$$U_{ijs} = \beta \ aesthetics_{ijs} + \beta \ compatibility_{ijs} + \ \beta \ durability_{ijs} + \beta \ out \ of \ pocket \ payment_{ijs} + \ \varepsilon_{ijs}$$

The utility appears as random, i.e., we cannot predict the choice. However, if we know the distribution of the random element, we can derive the probability of a choice. Depending on the assumptions for ε different analysis models must be applied. According to Bekker-Grob et al. [40], different restrictions have to be considered for each model. All analyses were performed using STATA software (version 15).

We **first** estimated a **conditional logit model (CLM)** to analyze how attributes determine the treatment choice. Basis for this analysis are constant choice sets per individual with varying attributes levels across alternatives as descriptive variable [41]. The CLM accounts for observed preference heterogeneity by including participants' characteristics (Z variables) [25]. Assuming that the utility of each alternative depends on its attributes, CLM models the influence of attributes that vary between alternatives on the selection probability regardless of alternatives (A, B, or opt-out) [42]. In addition, interactions in participants' (socioeconomic) characteristics can be considered. However, the CLM has some restrictions: it does not account for unobserved heterogeneity resulting from differences in preferences among participants with the same characteristics or random choices [25]. It models the choice between alternatives as a function of the alternatives' attributes but not of characteristics of the person making the choice [41]. Furthermore, CLM is making strong assumptions, i.e., independence of single, and irrelevant alternatives [Independence of Irrelevant Alternatives (IIA)] [36, 43]. The IIA describes the ratio of choice probability of two alternatives unaware of other alternatives. However, some alternatives vary more, and some are more similar in the choice sets of our study due to its design. These assumptions must be considered when data has panel character, e.g., participants making multiple choices [44], as it is the case in our study. These model characteristics and disadvantages were countered using other models. For CLM analysis the *clogit* command was used [45].

**Second**, we considered a **mixed logit model (MXL)** working with random parameters that vary between individuals to circumvent the IIA. The MXL allows for an estimation in which the independent assumption is violated by assuming that there is no independence in the choice behavior due to multiple choices by individual participants [25, 46]. It considers the alternatives' attributes and characteristics of the individuals [41]. The MXL estimates a distribution around each mean preference parameter to avoid potential bias in the estimated mean preference weights due to unobserved heterogeneity [47]. In our calculations, we did not include participant characteristics in the explainable component V but used the MXL to

estimate random parameters. This allowed us to account for random variation across participants, i.e., heterogeneity, of unobserved participant characteristics [48]. Random heterogeneity is evident from significant standard deviations per model parameter [49]. The calculations were performed using the *mixlogit* command [50].

**Third**, we estimated a **generalized multinomial logit model (G-MNL)** developed by Fiebig et al. [51]. This model provides for more flexible distributions, and accounts for unobserved preference heterogeneity by including random parameters into calculation, as well as scale heterogeneity [52]. Scale heterogeneity implies that choice behavior is more random for some individuals than for others [53]. Results of the G-MNL must be interpreted to allow for a more flexible distribution of confounded preference and scale heterogeneity, rather than estimating scale separately [25]. For G-MNL analysis we used the *gmnl* command [53].

Quality of all models was assessed using Akaike's and Schwarz' Bayesian information criterion (AIC and BIC) (and LL–log likelihood) [25]. The AIC estimates the amount of lost information of a model, and the BIC additionally adapts to the sample size [54]. AIC and BIC should therefore be as low as possible [55]. The most suitable model was chosen.

Based on the results of that model the **WTP** was calculated. WTP represents the amount of a cost attribute an average participant is willing to pay for one unit of an attribute in relation to the reference level [56]. In these linear models where each attribute in the utility function is associated with a single weight, the ratio of the two utility parameters was used to estimate the WTP. The following function calculates the participants' WTP, where $β_{ia}$ is a coefficient on one focused attribute "a", and $β_{ib}$ is a coefficient on the cost attribute "b" [57, 58], which is out-of-pocket payment in our study:

$$WTP = ß_{ia}/ß_{ib}$$

**Furthermore**, an estimation on alternative specific constants (ASC) was done via an **ASC-logit model (ASCL)** allowing us to include the individual characteristics as independent variables in the analysis. The aim was to examine the influence of regulatory instruments, participants' (socioeconomic) characteristics, and the importance of attributes on choice for or against a treatment at all ("treatment" vs. opt-out), and a SHI+ treatment. For the latter analysis, only choice sets representing exactly these treatments as a combination of levels were considered. Therefore, the number of choices is lower here. For analysis we used the *asclogit* command [59].

**Regression analyses** were performed to determine the influence of participants' socioeconomic characteristics [age, gender, income, employment status, residence (urban or rural)], treatment attributes, a bonus booklet, supplementary dental insurance, and its combination on the decision variables individual WTPmax, frequency of choosing opt-out and a SHI+ treatment. Analyses were conducted as follows: correlation analysis for refinement of subsequent multiple regression analysis, and graphical presentation of relevant categorial variables.

## Results

### Response rate and participants' characteristics

We received n = 762 questionnaires ensuring the response rate of r = 7.1%. According to our in-/exclusion criteria, data sets had to be excluded from further consideration. **Fig 1** gives an overview on numbers of selected questionnaires and data sets regarding criteria. Finally, n = 380 data sets could be included in the analysis. We were thus above the minimum required number of participants (see chapt. 2.2). Most of the participants belong to age group "50 to 59 years" (n = 103, 27.1%). The majority is female (n = 249, 65.5%), has a university degree (n = 166, 43.7%), and is employed full-time (n = 173, 45.5%). Medium and low household

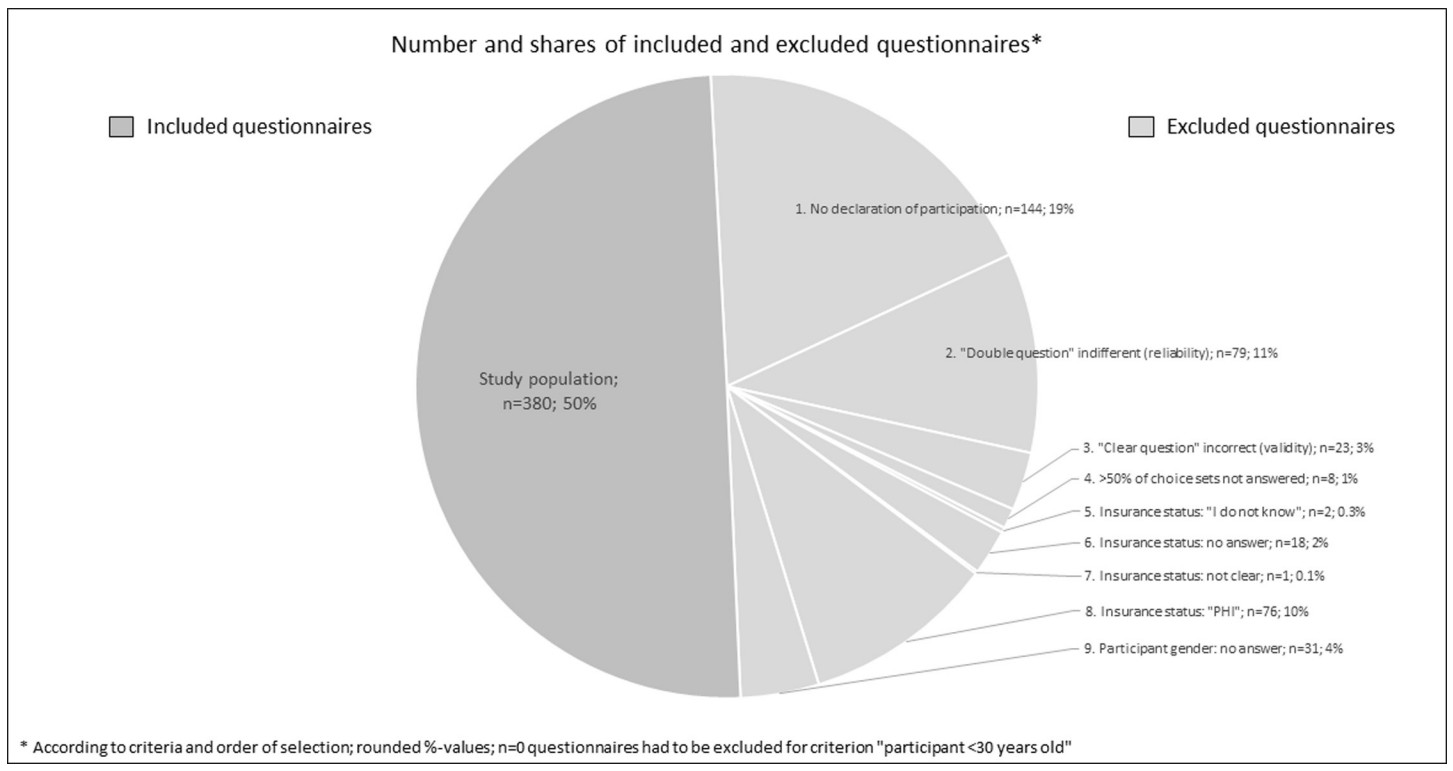

**Fig 1. Selection of questionnaires and data sets, numbers regarding citeria.**

incomes are most common (**Table 3**). Most of the participants indicated not having a dental supplementary insurance (n = 256, 67.4%). In contrast, a large proportion of our participants indicated having a bonus booklet (n = 329, 86.6%). Regarding their oral health, a large proportion of the participants stated to have a "good" (n = 170, 44.7%) or "very good" (n = 34, 9.0%) self-perceived status. Some participants (n = 49, 12.9%) had already decided against dental crown treatment in the past due to high costs (n = 17, 36.2%), or they considered a dental crown treatment as "unnecessary" (n = 14, 29.8%). Further reasons include questioning tooth preservation, and allergies (S3 File: Participants' reasons against a dental crown). Aditionally, majority of the participants (n = 238, 62.6%) indicated that they would always decide against a strong visible dental crown. Only a few would choose darker colors (golden-metal: n = 23, 29.9%; dark grey metallic: n = 6, 7.8%) (S4 File: Participants' decision for dental crown color). On average, it took the participants 18.5 minutes (range: 4–90min) to complete the questionnaire (S3 Table: Results on questions–Questionnaire Part C; S5 File: Codebook of analysis).

## Importance of treatment attributes

Regarding the importance of the four treatment attributes, durability (PT: n = 274, 72.1%; AT: n = 263, 69.2%) and compatibility (PT: n = 197, 51.8%; AT: n = 216, 56.8%) are assessed "very important" by our participants. Also, assessment of out-of-pocket payment is equally given for both teeth areas. The picture changes for the assessment of aesthetics. For PT aesthetics was assessed least as "very important" (n = 26, 6.8%) by our participants, and for AT it is most frequently given (n = 290, 76.3%) (S6 File: Importance of treatment attributes assessed by participants).

**Table 3. Participants' characteristics.**

| Participants' characteristics | | Total n = 380 (100%) |
|---|---|---|
| **Age** | 30 to 39 years | 68 (17.89) |
| | 40 to 49 years | 51 (13.42) |
| | 50 to 59 years | 103 (27.11) |
| | 60 to 69 years | 75 (19.74) |
| | 70 to 79 years | 43 (11.32) |
| | 80 to 89 years | 18 (4.74) |
| | 90 years and older | 1 (0.26) |
| | no answer / not clear | 21 (5.53) |
| **Gender** | Female | 249 (65.53) |
| | Male | 130 (34.21) |
| | Other | 1 (0.26) |
| **Residence** | urban region | 203 (53.42) |
| | rural region | 177 (46.58) |
| **Education** | (technical) university degree | 166 (43.68) |
| | vocational training | 115 (30.26) |
| | (technical) A-level | 22 (5.79) |
| | high school diploma– 10 years | 26 (6.84) |
| | high school diploma– 9 years | 13 (3.42) |
| | Other | 3 (0.79) |
| | no answer / not clear | 35 (9.21) |
| **Employment** | full-time | 173 (45.53) |
| | part-time | 54 (14.21) |
| | university/college student (not employed) | 1 (0.26) |
| | Unemployed | 11 (2.89) |
| | retirement due to illness | 16 (4.21) |
| | retirement due to age | 95 (25.00) |
| | Other | 13 (3.42) |
| | no answer / not clear | 17 (4.47) |
| **Income (household/month)** | under 500 € | 8 (2.11) |
| | 500 to under 1,500 € | 48 (12.63) |
| | 1,500 to under 2,500 € | 124 (32.63) |
| | 2,500 to under 4,500 € | 152 (40.00) |
| | 4,500 to under 6,500 € | 48 (12.63) |
| | over 6,500 € | 6 (1.58) |
| | no answer / not clear | 26 (6.84) |

## Results of discrete-choice, willingness-to-pay, and regression analysis

Estimations using the different models have produced different AIC and BIC. Lowest coefficient values were calculated for the MXL model (AIC/BIC PT: 5,176/5,312; AT: 4,692/4,827). Since MXL also makes the realistic acceptable assumption of including random parameters, we considered these results. For this reason, we have also performed the WTP calculation based on the MXL using the *wtp* command [57]. The "no. of observations" in the (appendix) tables do not refer to the population sample size, but to the dataset rows included in the analyses, and therefore vary between the models. Rows with missings, and of non-relevant choice sets (e.g., SHI levels in ASCL) were excluded. Single G-MNL results are presented in the following (S4 Table: Coefficients of G-MNL estimations, including marginal effects; S5 Table: Coefficients of CLM estimations, including marginal effects).

**(i.) Discrete-choice-models.** The participants preferred lightly visible (Coef.: 0.687, p<0.01) and natural color PT (Coef.: 1.290, p<0.01) compared to a strongly visible color of the teeth. The probability of choosing natural color instead of lightly visible teeth is almost four times larger compared to the reference level, measured by marginal effects [dy/dx: 1.247, p<0.01; vs. lightly visible (dy/dx: 0.334, p<0.01)]. In addition, the participants preferred a treatment without risk of incompatibility (Coef.: 0.465, p<0.01). A durability of 25 years was associated with higher preferences by our participants compared to other levels [Coef.: 1.540, p<0.01; e.g., vs. 15 years (Coef.: 0.477, p<0.01)]. The participants preferred low out-of-pocket payments, e.g., 50 € (Coef.: 0.776, p<0.01) and 150 € (Coef.: 0.965, p<0.01).

The results are similar for AT. Lightly visible (Coef.: 1.026, p<0.01) and natural color teeth (Coef.: 3.392, p<0.01) are assigned higher preferences than compared to strongly visible crowns. Similarly, no-risk treatments regarding compatibility (Coef.: 0.200, p<0.05) are assigned higher preferences by the participants, as well as for the longest durability of 25 years (Coef.: 0.835, p<0.01).

Comparing the results of both teeth areas, aesthetics of the AT can be considered more important for the participants since the probability of choosing a natural color crown is approximately three times higher than for the PT compared to the reference level according to marginal effects results [dy/dx: 3.449, p<0.01; vs. PT (dy/dx: 1.247, p<0.01)]. For teeth in both teeth areas, no-risk treatments and the highest possible durability of 25 years are preferred by the participants. The preferred out-of-pocket payment corresponds to the co-payment of current SHI standard care, for PT and AT. Considering all coefficients, the level natural color for AT stands out. Overall, this level of the attribute aesthetics is preferred by the participants in their decision-making. Coefficients of the MXL model for PT can be seen in **Table 4** (S6 Table: Coefficients of MXL estimations for anterior teeth, S7 Table: Marginal effects of MXL estimations).

**(ii.) ASC-logit models.** *(ii.1) Analysis of choice between "treatment" and "no treatment (opt-out)".* For PT, the opt-out alternative, i.e., no treatment, was selected in 25.7% (n = 774) of the choice scenarios by the participants [AT: 37.2% (n = 1,122)]. A combination of bonus booklet and supplementary dental insurance increases the likelihood of choosing a treatment (Coef. PT: 0.335, AT: 0.377; p<0.05). This is also true for higher aged participants (Coef. PT: 0.119, AT: 0.087; p<0.01), being a resident of an urban region (Coef. PT: 0.131, AT: 0.129; p<0.01), and for an increased importance of the attribute aesthetics (Coef. PT: 0.154, p<0.01; AT: 0.1, p<0.05). Furthermore, gender, bonus booklet, and importance of out-of-pocket payments have an impact on our participants' choice behavior.

*(ii.2) Analysis of choice between "SHI standard care" and "treatment beyond SHI standard care (SHI+)".* In 31.3% (n = 480) of the choice scenarios for PT, in which a decision could be made between a SHI+ treatment versus treatment below standard care (n = 1,533 decisions in total), the participants chose SHI+. This occurred more frequently for AT: in 49.8% (n = 449) of the decisions (n = 902 in total) they decided for SHI+. As the importance of the attribute aesthetics increases, the participants decided against SHI standard care for teeth of both teeth areas and favored treatments beyond that (Coef. PT: -0.243, p<0.01; AT: -0.18, p<0.05). For AT, the participants chose SHI standard care as the importance of out-of-pocket payments increases (Coef.: 0.17, p<0.1). Regression analysis was additionally performed with the same dependent variables (see chapt. 3.1.iv.) [S8 Table: Coefficients of ASCL estimations "treatment" vs. "no treatment (opt-out)", S9 Table: Coefficients of ASCL estimations "SHI standard care" and "treatment beyond SHI standard care (SHI+)"].

**(iii.) Willingness-to-pay.** For PT, the out-of-pocket payment of SHI standard care is set at 150 €. For the attributes aesthetics (strongly visible) and compatibility (risk) levels, there is no participants' willingness-to-pay. Regarding a durability of 15 years, WTP is higher (258 €).

**Table 4. Coefficients of the MXL model.**

| Mixed logit model (MXL) | | | | | | | | |
|---|---|---|---|---|---|---|---|---|
| **Posterior teeth** | | | | | | | | |
| **Attributes (Ref. *negative* levels)** | **Levels** | **Coef.** | **Std. Err.** | **t-value (z)** | **p-value (P>|z|)** | **[95% Conf. interval]** | | **Sig.** |
| Aesthetics | *strongly visible–reference level* | | | | | | | |
| | lightly visible | 0.687 | 0.103 | 6.680 | 0.000 | 0.485 | 0.888 | *** |
| | natural color | 1.290 | 0.113 | 11.410 | 0.000 | 1.068 | 1.512 | *** |
| Compatibility | *1 out of 10,000 people with allergic or local toxic reaction–reference level* | | | | | | | |
| | no risk | 0.465 | 0.086 | 5.400 | 0.000 | 0.296 | 0.634 | *** |
| Durability | *5 years–reference level* | | | | | | | |
| | 10 years | 0.159 | 0.121 | 1.310 | 0.189 | -0.078 | 0.395 | |
| | 15 years | 0.477 | 0.112 | 4.270 | 0.000 | 0.258 | 0.695 | *** |
| | 25 years | 1.540 | 0.140 | 11.000 | 0.000 | 1.266 | 1.815 | *** |
| Out-of-pocket payment | *600 €–reference level* | | | | | | | |
| | 450 € | 0.191 | 0.113 | 1.690 | 0.091 | -0.030 | 0.411 | * |
| | 150 € | 0.965 | 0.107 | 9.000 | 0.000 | 0.755 | 1.175 | *** |
| | 50 € | 0.776 | 0.130 | 5.960 | 0.000 | 0.521 | 1.031 | *** |
| **Log likelihood** | -2,569.2494 (Iteration 8) | | | | | | | |
| **Prob > chi2** | 0.0 | | | | | | | |
| **LR chi2(9)** | 459.8 | | | | | | | |
| **No. of observations** | 9,039 | | | | | | | |

AIC / BIC (Akaike's & Schwarz's Bayesian information criteria): 5,176 / 5,312

*** p < .01,

** p < .05,

* p < .1

Considering attributes' most positive "high quality" treatment levels, we see the following: for natural color teeth, the WTP is 380 €, for a treatment without risk of incompatibility, the participants are willing to pay 162 €, and for dental crowns with 25 years durability 508 €. For AT, the out-of-pocket payment is 200 €. The participants would pay 362 € for lightly visible anterior teeth, but there is no willingness-to-pay for a risk of incompatibility. For dental crowns with a durability of 10 years, the participants are willing to pay 73 €, for natural color teeth 914 €, for risk-free treatments 92 €, and 282 € for a durability of 25 years. Comparing both teeth areas, we see that WTP for aesthetics is higher for anterior teeth, especially at the natural color level. On the contrary, WTP for compatibility and durability is higher for PT (S10 Table: WTP analysis framework, and results).

**(iv.) Regression analysis.** The results of correlation analyses, and calculation of the variance inflation factors (VIF), led to the exclusion of certain variables (e.g., combination of bonus booklet and dental supplementary insurance) from regression analyses. Statistical significance (applicable to the following reported results) was not given for all calculations.

With an increasing age (Coef. PT: -13.78, p<0.01; AT: -13.73, p<0.05), WTPmax decreases. Furthermore, with the presence of a bonus booklet (Coef. PT: 114.79, p<0.01; AT: 111.66, p<0.01), it increases. For AT, also gender (Coef. female: -91.01, p<0.01) has an influence. Female participants more often accepted high out-of-pocket payment amounts.

Individual variables are also correlated with the decision against a treatment ("no treatment") for PT and AT: with increasing age (Coef. PT: 0.21, p<0.01; AT: 0.21, p<0.01), residence in smaller towns (Coef. PT: -0.24, p<0.05; AT: -0.25, p<0.05), non-existence of a bonus booklet (Coef. PT: -1.21, p<0.05; AT: -1.21, p<0.05), and having a dental supplementary

insurance (Coef. PT: 0.71, p<0.05; AT: 0.71, p<0.05), the decision against a treatment has been made more frequently. The participants' gender (Coef. female: 0.81, p<0.05) also plays a role regarding decisions for AT. Female respondents are less likely to decide against a treatment.

The more important co-payment (Coef. PT: -0.21, p<0.05; AT: -0.21, p<0.1), the less often a treatment outside SHI standard care was chosen. With the existence of a bonus booklet (Coef. PT: 0.96, p<0.01; AT: 0.74, p<0.05), SHI+ was chosen more often. For PT, the importance of aesthetics (Coef. 0.26, p<0.01) also has an influence. The more important aesthetics, the more often SHI+ was chosen by the participants. For AT, gender (Coef. female: -0.51, p<0.05), and residence (Coef.: 0.17, p<0.05) additionally influenced their decisions. Females and residents of smaller towns were less likely to choose a more cost-intensive treatment (S11 Table: Tables on correlation analysis, VIF values, and regression analysis; S7 File: Regression plots on WTPmax and SHI+ analysis).

## Discussion

This study provides important insights into factors determining patients' choice behavior in dental care, while distinguishing between the two teeth areas, PT and AT. The focus of the choice analyses was on highest benefit expectations assigned by the participants to attributes and its levels of dental crown treatment, as well as the participants' willingness-to-pay. Further analyses focused on incentive measures provided by SHI and private health insurance, on choice for or against a treatment ("no treatment"), and for a treatment beyond SHI standard care, and the influence of the participants' (socioeconomic) characteristics in decision-making.

Our results show that aesthetics is an important factor for the participants in their choice of a dental crown treatment. For AT, aesthetics has a higher weight for the participants. Highest benefit allocations are assigned to "natural color", i.e., tooth-colored, dental crowns, which should be indistinguishable from natural teeth in terms of visibility. Results on the importance of aesthetics underline our choice analysis estimates. Furthermore, the importance of aesthetic aspects of AT has already been shown in previous research [60, 61]. For PT, durability and treatment attributes such as functionality [62, 63] might be more meaningful. Nevertheless, even for PT, natural color teeth are preferred over strongly visible. Risk of a local toxic or allergic reaction seems to have rather less weight among the participants. For PT and AT, the coefficient for non-risk in the choice analysis is small. It can be assumed patients accept those risks. These values may result from the experimental design, i.e., extremely preferred (especially for AT) or non-preferred expressions were opposite to the risk attribute level. Besides, this may result from the fact that the probability of occurrence of a local toxic or allergic reaction appears low, also based on the participants' awareness and experiences (e.g., does not know about allergies, former allergic reactions were mild) [64]. The attribute durability of a dental crown has a great influence on the participants' decision-making, for both teeth areas. Highest benefit is clearly assigned to the highest duration of 25 years. A long life cycle could mean convenience for patients: lower costs in the long term, fewer visits at the dentist which may be painful, etc. However, present conditions may stand in the way of this patient desire. Dental crowns made of common materials (e.g., SHI standard care restorations) have an average life span of 15 years. For high-quality and -priced dental crowns, durability could be a few years higher (e.g., gold alloys) [65, 66]. Ultimately, the quest for long lifetime materials needs to be realized through further research activities. Out-of-pocket payments play an important role in our participants treatment choice, independently from teeth area. Nevertheless, these would require high co-payments, especially for AT. In the context of choice analysis, it is

important to note that this is a benefit allocation. The participants might have associated high costs with other treatment characteristics, such as high quality [67].

We determined the participants' willingness-to-pay for treatments with a natural color dental crown, i.e., the best possible attribute level. For both teeth areas and both cost attributes, the maximum amount to pay is above the level of SHI standard care. The participants are willing to pay much more for AT. High WTP values are possible due to the design with closed-ended questions [68]. Nevertheless, it should be noted that these values refer to the entirety of the participants and cannot be attributed to an individual participant [34]. Further analysis of the cost attribute revealed that willingness-to-pay per participants decreases with increasing age. Reduced incomes at an older age (e.g., pension), expensive treatments, or a reduced awareness of aesthetically high-quality as well as prioritization of functionality could be a reason [69–71]. Presumably, the older the patient, the more intentional the dentist is in communicating that a dental crown is possibly the last and only alternative for tooth preservation [72]. There may be financial and pragmatic reasons for choosing the SHI standard care alternative. Previous studies have examined inequalities in dental treatment utilization. Besides income, financial wealth is one reason [73–75].

Descriptive analysis showed that a large proportion of the participants owns a bonus booklet, but only a few participants have taken out a dental supplementary insurance. The proportions correspond to those for Germany: only about a quarter of SHI-insured people have private supplementary insurance [76]. Some participants stated they had already rejected dental crowns in the past, for reasons of cost and lack of necessity from their point of view. Combining bonus booklet and supplementary insurance makes patients more likely to choose a treatment than no treatment at all, for both teeth areas. It should be noted that this approach is used to reduce out-of-pocket payments, regardless of which form of care is chosen. However, it should also be noted that private supplementary insurances incur fees and cannot be financed by every patient [77]. If no dental supplementary insurance has made, out-of-pocket payment might remain at a high level. Patients might choose the most inexpensive alternative, including no treatment, although, SHI provides incentive measures. Medical necessity of dental treatments seems to be irrelevant for the participants. One assumption of our questionnaire was, that the dental crown treatment is found to be necessary by a dentist. Patients' attitude has been reported in former articles [78, 79]. Overall, SHI standard care is accepted by patients, especially in older age groups, when aesthetics takes a back seat, and cost and functionality aspects become more relevant. However, it is apparent that there is a desire for more aesthetically pleasing and long-lasting alternatives. The former point is given especially for AT. Many patients keep a bonus booklet and make use of it (proof of at least 5 years annual check-ups in a row). SHI should target further possibilities and combinations of bonus measures to reduce access barriers to care and improve utilization of routine check-ups that can prevent caries. These measures could be linked to conditions that promote patients' oral health behavior, as the bonus booklet successfully demonstrates [80].

Some limitations must be mentioned. The study is limited to the states of Berlin and Brandenburg. A region-typical choice behavior is conceivable here. When selecting the regional areas to which the questionnaires were sent, we took care to ensure an equal distribution of household incomes across federal states' districts and counties. Nevertheless, a large proportion of the participants tended to belong to low household incomes groups. This may have biased the results, particularly regarding financial preferences. Since the sample is small due to the experimental design, descriptive results may not be representative for Germany. In the results of the models, especially ASCL, there is partly no statistical significance, although we have reached the minimum sample size. Accordingly, there are gaps in the answers to the research questions. Although the treatment attributes and their levels were designed to be as

realistic as possible, it should be noted that the presentation of the treatment alternatives in the questionnaire probably does not reflect real life decision-making situations of the individual participants: choice scenarios were limited to a few attributes and levels. More factors probably play a role in patients' treatment decision, including possible medical consequences or the relationship with the dentist [81, 82], and there may be more than two alternatives to choose from. The questionnaire, including n = 18 choice scenarios, was also very complex. Possibly, participants applied heuristics to simplify decision situations [1]. Also, conditions under which the questionnaires were completed are unclear, e.g., maybe participants were influenced by relatives. Additionally, the participants could not ask questions in case of any ambiguities.

The often non-statistical-significance of the regression analyses results can be explained by the fact that the participant number was small. However, there is no guideline for minimum population numbers for regressions. Study's focus was a DCE, in which the experimental design allows small populations [57]. Statistical significances were given for most results in the choice analyses. It should also be noted that in the choice analyses, values of the coefficients were sometimes small, i.e., close to the reference level, or close between levels. Results should be interpreted accordingly.

## Conclusion

Dental interventions such as crown treatments, involve difficult decisions on the optimal allocation of resources for health care systems and patients. This study provides important insights into patient preferences for crown treatment in Germany. Findings show that aesthetic for AT and PT as well as out-of-pocket payments for PT play an important role in the decision for dental crown treatments. Overall, participants are willing to pay more out of pocket compared to out-of-pocket payment that arises for SHI standard care, with a considerably higher willingness-to-pay for AT. Having a bonus booklet increased the willingness-to-pay. Although, the findings should be interpreted with caution due to limitations of choice experiments and the regional restriction to two federal states in Germany, it may also be valuable for policy makers and health insurance funds in developing dental health care programs, creating incentive structures, and planning the provision of dental services that better match patient preferences. For further studies, participants from all income groups should be targeted and included in the analysis in equal proportions (e.g., randomization within income groups), and regression analyses should be planned with larger populations.

## Supporting information

**S1 File. Questionnaire.**
(PDF)

**S2 File. Document "Aesthetics".**
(PDF)

**S3 File. Participants' reasons against a dental crown.**
(TIFF)

**S4 File. Participants' decision for dental crown color.**
(TIFF)

**S5 File. Codebook of analysis.**
(XLSX)

**S6 File. Importance of treatment attributes assessed by participants.**
(TIF)

**S7 File. Regression plots on WTPmax and SHI+ analysis.**
(DOCX)

**S1 Table. Design output.**
(DOCX)

**S2 Table. Calculation of D-efficiency.**
(DOCX)

**S3 Table. Results on questions–Questionnaire Part C.**
(DOCX)

**S4 Table. Coefficients of G-MNL estimations, including marginal effects.**
(DOCX)

**S5 Table. Coefficients of CLM estimations, including marginal effects.**
(DOCX)

**S6 Table. Coefficients of MXL estimations for anterior teeth.**
(DOCX)

**S7 Table. Marginal effects of MXL estimations.**
(DOCX)

**S8 Table. Coefficients of ASCL estimations "treatment" vs. "no treatment (opt-out)".**
(DOCX)

**S9 Table. Coefficients of ASCL estimations "SHI standard care" and "treatment beyond SHI standard care (SHI+)".**
(DOCX)

**S10 Table. WTP analysis framework, and results.**
(DOCX)

**S11 Table. Tables on correlation analysis, VIF values, and regression analysis.**
(DOCX)

**S1 Dataset.**
(XLSX)

## Acknowledgments

For a better readability in the text, we used the terms posterior and anterior teeth. In the questionnaire (according to SHI context) the terms "non-visible tooth area" and "visible tooth area" were used representing different tooth numbers from a definitional point of view (see dental scheme in S1 and S2 Files).

We sincerely thank the respondents for participating in our study. Also, we thank the illustrator Juliane Lüke for a layout of the extra document "Aesthetics", and student assistant Hauke Langhoff.

We acknowledge support by the German Research Foundation and the Open Access Publication Fund of TU Berlin.

## Author Contributions

**Conceptualization:** Susanne Felgner.

**Data curation:** Susanne Felgner.

**Formal analysis:** Susanne Felgner.

**Funding acquisition:** Cornelia Henschke.

**Investigation:** Susanne Felgner.

**Methodology:** Susanne Felgner.

**Project administration:** Cornelia Henschke.

**Software:** Susanne Felgner.

**Supervision:** Cornelia Henschke.

**Validation:** Susanne Felgner.

**Visualization:** Susanne Felgner.

**Writing – original draft:** Susanne Felgner.

**Writing – review & editing:** Cornelia Henschke.

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
