## [Decision Letter · Decision Letter 0]

8 Aug 2022

PONE-D-22-18370A discrete-choice experiment and an analysis of patients' willingness-to-pay in dental carePLOS ONE

Dear Dr. Felgner,

Thank you for submitting your manuscript to PLOS ONE. After careful consideration, we feel that it has merit but does not fully meet PLOS ONE’s publication criteria as it currently stands. Therefore, we invite you to submit a revised version of the manuscript that addresses the points raised during the review process.

The referee, who is an expert in the field, makes some excellent and constructive points. Please make sure to address each of them when preparing the revised version.

We look forward to receiving your revised manuscript.

Kind regards,

Ted Loch-Temzelides

Academic Editor

PLOS ONE

Journal Requirements:

Additional Editor Comments:

This paper deals with a topical health choice question in the context of dental care. It investigates how treatment attributes and out-of-pocket expenses determine patients’ treatment choices. The statistical analysis is based on a data set created by the authors involving responses to questionnaires. The analysis reveals that aesthetics and durability play an important role in patients’ decisions. In addition, out-of-pocket cost aspects appear to be at least somewhat relevant in the choice of dental care services.

The analysis is well-executed and tightly focused. When revising the paper, the authors should spend some additional time motivating the particular statistical techniques used and why they are the appropriate methods to address the problem at hand.

Line 491: the last sentence in the conclusion appears to be incomplete.

Reviewers' comments:

Reviewer's Responses to Questions

**Comments to the Author**

1. Is the manuscript technically sound, and do the data support the conclusions?

Reviewer #1: Partly

2. Has the statistical analysis been performed appropriately and rigorously? 

Reviewer #1: Yes

3. Have the authors made all data underlying the findings in their manuscript fully available?

Reviewer #1: No

4. Is the manuscript presented in an intelligible fashion and written in standard English?

Reviewer #1: Yes

5. Review Comments to the Author

Reviewer #1: Referee Report

A discrete-choice experiment and an analysis of patients' willingness-to-pay in dental care :A discrete-choice experiment in dental care

This paper uses a discrete choice experiment to investigate the effect of treatment attributes on patients’ treatment choice of dental crowns and whether out-of-pocket payments represent a barrier to access dental care. The authors analyzed willingness to pay and how socioeconomic characteristics affect it. Out of 10,752 emailed questionnaires and 762 returns, the authors include 380 in the analysis and find that aesthetics and durability of the crowns are the more preferred attributes, and that demographic characteristics influenced the willingness to pay. This paper is quite interesting. The research design, analysis, and discussion of potential policy concerns was quite clear and concise. I have a few comments.

Comments on the content

1. It would be great to focus on the novelty of the paper. I realize that the authors mention that they focus on a treatment that is found in the SHI benefit basket but a discussion of why it is important and how it is specifically differentiated from previous studies would improve readability substantially.

2. In Line 89, the authors mention the methods used. I agree that these methods are well understood in the health sciences literature but some discussion on why these are relevant to answer the question of interest would be great.

3. The experiment provides two unlabeled alternatives and the option of no treatment. However, as the authors note, individuals face multiple choices. Therefore, just providing two choices may not elicit the preferences as there might be behavioral biases in real life. For example, patients might choose a worse alternative in all attributes if they are averse to searching for multiple treatments or if that was marketed better. In this context, it would have been interesting to see if patients respond differently to different size of choice sets.

4. There could be a selection bias in the response as most individuals included in the study have medium and low household incomes. Hence, the estimated parameters could be biased. Specifically, the authors mention earlier in the text that dental crown treatments might require significant financial resources but since most of the individuals who responded have low household incomes, it raises questions about the external validity of the experiment.

5. I am quite confused by the different sample size on the choice model result tables. It would be good to explain why these sample sizes are different.

6. The lack of statistical significance on the results for anterior teeth maybe due to a low sample size. It would be good to send more questionnaires. If not, then it would improve readability to move that table to the appendix.

7. It is great that the authors discuss the external validity of their experiment, but it would have been interesting to get their take on why that might be a problem in the specific case of dental crowns.

8. It would be interesting to see how the choice behavior is dependent on the accessibility to dental clinics by the respondent at an administrative district level as that data

Other Minor Comments

1. I found some typos in the text that I mention below.

a. Line 54- “methods” instead of “methodes”

b. Line 73 – “decision making” instead of “decisions making”

c. Line 81 – “addressing” instead of “adressing”

d. Line 119 – “scenarios” instead of “szenarios”

e. Line 155 - “backgrounds” instead of “backrounds”

f. Line 246 – “Akaike” instead of “Akaik”

g. Line 303 – “Coefficient” instead of “coefficiant” (Recurring Mistake)

h. Line 383 – “Booklet” instead of “booklet booklet”

2. I also found that there were some formatting errors with the quotes in some places. Rather than having ‘ “text” ‘, I found some places with ‘ ,,text” ‘.

6. PLOS authors have the option to publish the peer review history of their article (what does this mean?). If published, this will include your full peer review and any attached files.

Reviewer #1: No

---

## [Author Response · Author response to Decision Letter 0]

27 Dec 2022

Dear Prof. Dr. Loch-Temzelides,

First, we would like to thank you and your team for giving us the opportunity to revise the manuscript. We addressed all mentioned concerns carefully and have responded to each recommendation as directly as possible. We have found your and the reviewer’s comments to be very insightful and helpful and feel that the manuscript has greatly benefited from it.

Below please find our responses (in non-bold letters) to each of the points raised by you, including the points on journal requirements, and the reviewer (in bold letters). 

We have included a marked-up copy of our manuscript highlighting the changes to the original version ("Revised Manuscript with Track Changes", revised parts are marked in yellow) as well as an unmarked version ("Manuscript").

Furthermore, we would like to politely ask for a change of the manuscript title from "A discrete-choice experiment and an analysis of patients' willingness-to-pay in dental care" to "Patients' preferences in dental care: a discrete-choice experiment and an analysis of willingness-to-pay". The new title more clearly reflects the contents of the manuscript. This will give the (potential) reader more information about the manuscript and should arouse interest even more.

Please do not hesitate to contact me if anything remains unclear or further revisions are needed. 

Once again, we thank you very much for your time! 

Yours sincerely,

Susanne Felgner

Journal Requirements

Thank you for the advice on formatting. We have removed the short title on the title page and made revisions in the manuscript, i.e., we changed “Figure 1” to “Fig 1” (line 305); and we changed level 1, 2 and 3 (and 4) headings to 18pt, 16pt and 14pt (and 12pt) font, and further changed headings to bold type, and removed italics.

In agreement with the ethics committee and our institution's data protection officer we have added an anonymized minimal data set (“Minimal_data_set”) of the choice analysis, without considering individual participants’ characteristics, to the Supporting Information of our manuscript. Individual participant data may only be presented in aggregated form according to the ethics application, as in Table 3 on participants’ characteristics (p.12f.). We have changed the information in the data availability statement accordingly. 

The ethics application has been accepted by the ethics committee of the Charité Universitätsmedizin Berlin (application no. EA4/109/19; contact details: Charité – Universitätsmedizin Berlin, Ethikkommission der Charité, Charitéplatz 1, 10117 Berlin). 

In addition to the authors, only the employees of the department have access to the data.

Additional Editor Comments

This paper deals with a topical health choice question in the context of dental care. It investigates how treatment attributes and out-of-pocket expenses determine patients’ treatment choices. The statistical analysis is based on a data set created by the authors involving responses to questionnaires. The analysis reveals that aesthetics and durability play an important role in patients’ decisions. In addition, out-of-pocket cost aspects appear to be at least somewhat relevant in the choice of dental care services.

Thank you for the assessment.

The analysis is well-executed and tightly focused. When revising the paper, the authors should spend some additional time motivating the particular statistical techniques used and why they are the appropriate methods to address the problem at hand.

Thank you for the advice. The statistical methods used in our study are choice analysis (i.e., discrete-choice experiment analysis), willingness-to-pay analysis (WTP), and regression analysis. Choice analysis was chosen because it can be used to elicit respondents' preferences. It is used particularly in health sciences to explain patients' choice behavior. The different models applied resulted from the structure of the data, which resulted from the assumptions of the scenarios. For example, we decided for an unlabeled design of the choice sets, since in real-world patients are free to choose between various treatment alternatives. As it was not our concern to investigate or improve an already existing treatment of defined characteristics, a labeled design was excluded for the experiment. The unlabeled design, as well as the number of choice alternatives, lead us to the application of the different models to fulfill our study’s aims: (1) conditional logit model (CLM) to consider treatment characteristics for participants’ decision-making; (2) mixed logit model (MXL) as CLM alternative to also consider treatment characteristics but using an estimate of random parameters to account for heterogeneity, i.e., random variation across participants, due to unobserved characteristics; (3) generalized multinomial logit model (G-MNL) considering scale heterogeneity across participants in this regard; and (4) ASC logit model (ASCL, estimations on alternative specific constants) to include the individual characteristics as independent variables on defined choice scenarios (treatment vs. no treatment, SHI+ treatment). The result values of considered models for analysis on treatment characteristics on the participants’ choice behavior had to meet certain criteria (Akaike's and Schwarz' Bayesian information criterion: AIC and BIC). A willingness-to-pay analysis directly followed the mixed logit model choice analysis, since best AIC and BIC values were given and due to the methodological reason that WTP estimations are based on choice analysis results. Using willingness-to-pay analysis enabled us to consider patients' treatment choices. It allowed us to determine respondents’ monetary maximum value for each attribute level. Thus, a comparison of the amount of out-of-pocket payments resulting from SHI’s benefit basket and respondents’ willingness-to-pay was possible, which in turn allowed for an assessment of SHI co-payment from patients’ perspective. We chose an additional regression analysis to calculate an assumed relationship between socioeconomic characteristics of the participants and variables expressing decisions (e.g., choosing opt-out). 

In the methods section, we now give a more detailed introduction why we used these methods in the study: “Discrete-choice experiments (DCE) are an established instrument particularly in health sciences [14] for measuring patient preferences in their choice behavior by estimating benefit assignments, and for calculating willingness-to-pay (WTP) taking out-of-pocket payments into account. Although, medical professionals usually recommend a treatment option, due to restricted coverage in oral health care services, final decisions are largely guided by patient preferences. A DCE is best suited to collect data for analyzing preferences of patients and their WTP. This is especially the case as patients have to decide between different treatment opportunities that come along with large variations in out-of-pocket payments. To analyze patient preferences, we therefore conducted a DCE. Furthermore, we analyzed overall (and individual) WTP to compare monetary value of respondents’ willingness-to-pay and the SHI out-of-pocket payment for each attribute. In addition, we conducted regression analyses to calculate the relationship between socio-economic and other characteristics of participants and defined decision variables. Descriptive analyses were used to illustrate quantitative results (S3, S4, and S6 Files).” (line 94ff.). Further, we revised the methods section to better explain and justify the choice and application of the MXL to the reader since to us this seemed to be not clear enough so far. Following information has been added: “The MXL estimates a distribution around each mean preference parameter to avoid potential bias in the estimated mean preference weights due to unobserved heterogeneity [47]. In our calculations, we did not include participant characteristics in the explainable component V but used the MXL to estimate random parameters. This allowed us to account for random variation across participants, i.e., heterogeneity, of unobserved participant characteristics [48]. Random heterogeneity is evident from significant standard deviations per model parameter [49].” (line 264ff.).

Line 491: the last sentence in the conclusion appears to be incomplete.

Thank you for pointing out this. We have changed the sentence in the conclusion accordingly: "For further studies, participants from all income groups should be targeted and included in the analysis in equal proportions (e.g., randomization within income groups), and regression analyses should be planned with larger populations." (line 529ff.).

Reviewers’ comments to the author

I Reviewer's Responses to Questions

1. Is the manuscript technically sound, and do the data support the conclusions?

The manuscript must describe a technically sound piece of scientific research with data that supports the conclusions. Experiments must have been conducted rigorously, with appropriate controls, replication, and sample sizes. The conclusions must be drawn appropriately based on the data presented. Reviewer #1: Partly

Several measures were applied to make the experiment rigorous: (1) a d-efficient design was applied (line 123). The d-efficiency value was calculated and an acceptable high value of 100% was achieved (S2 Table), and (2) SAS software was used to design the choice sets so that bias and human error could be excluded (line 127f.). Further, (3) a pretest was conducted (line 178ff.). In this context, the comprehensibility of the questionnaire was reviewed and revised. 

You are quite right that conclusions should be drawn more clearly from the results. We therefore revised the conclusions section in the text: “Dental interventions such as crown treatments, involve difficult decisions on the optimal allocation of resources for health care systems and patients. This study provides important insights into patient preferences for crown treatment in Germany. Findings show that aesthetic for AT and PT as well as out-of-pocket payments for PT play an important role in the decision for dental crown treatments. Overall, participants are willing to pay more out of pocket compared to out-of-pocket payment that arises for SHI standard care, with a considerably higher willingness-to-pay for AT. Having a bonus booklet increased the willingness-to-pay. Although, the findings should be interpreted with caution due to limitations of choice experiments and the regional restriction to two federal states in Germany, it may also be valuable for policy makers and health insurance funds in developing dental health care programs, creating incentive structures, and planning the provision of dental services that better match patient preferences.” (line 519ff.). 

Accordingly, we revised the abstract conclusion: “This study provides important insights into patient preferences for dental crown treatment in Germany. For our participants, aesthetic for AT and PT as well as out-of-pocket payments for PT play an important role in decision-making. Overall, they are willing to pay more than the current out-of-pocket payments for what they consider to be better crown treatments. Findings may be valuable for policy makers in developing measures that better match patient preferences.” (line 43ff.).

2. Has the statistical analysis been performed appropriately and rigorously? Reviewer #1: Yes

Thank you for the assessment.

3. Have the authors made all data underlying the findings in their manuscript fully available?

The PLOS Data policy requires authors to make all data underlying the findings described in their manuscript fully available without restriction, with rare exception (please refer to the Data Availability Statement in the manuscript PDF file). The data should be provided as part of the manuscript or its supporting information, or deposited to a public repository. For example, in addition to summary statistics, the data points behind means, medians and variance measures should be available. If there are restrictions on publicly sharing data—e.g. participant privacy or use of data from a third party—those must be specified. Reviewer #1: No

We have added an anonymized minimal data set (“Minimal_data_set”) of the choice analysis, without considering individual participants’ characteristics, to the Supporting Information of our manuscript. This complies with the requirements of the ethics committee and our institution's data protection officer. Individual participant data may only be presented in aggregated form according to the ethics application, as in Table 3 on participants’ characteristics (p.12f.). We have changed the information in the data availability statement accordingly. Please excuse if our data availability statements have been incorrect and misleading so far!

4. Is the manuscript presented in an intelligible fashion and written in standard English?

Reviewer #1: Yes

Thank you for the assessment.

5. Review Comments to the Author

Reviewer #1: Referee Report - A discrete-choice experiment and an analysis of patients' willingness-to-pay in dental care :A discrete-choice experiment in dental care

This paper uses a discrete choice experiment to investigate the effect of treatment attributes on patients’ treatment choice of dental crowns and whether out-of-pocket payments represent a barrier to access dental care. The authors analyzed willingness to pay and how socioeconomic characteristics affect it. Out of 10,752 emailed questionnaires and 762 returns, the authors include 380 in the analysis and find that aesthetics and durability of the crowns are the more preferred attributes, and that demographic characteristics influenced the willingness to pay. This paper is quite interesting. The research design, analysis, and discussion of potential policy concerns was quite clear and concise. I have a few comments.

Thank you for the assessment.

II Comments on the content

1. It would be great to focus on the novelty of the paper. I realize that the authors mention that they focus on a treatment that is found in the SHI benefit basket but a discussion of why it is important and how it is specifically differentiated from previous studies would improve readability substantially.

Thank you for your suggestion. As described in the introduction of the manuscript, we focus on full dental crowns, a treatment that is included in the SHI benefit basket. The special feature of this treatment is that the attributes out-of-pocket payment and aesthetics can vary greatly, depending on the chosen treatment. The assumed most (un)attractive treatments for patients according to attribute levels may be in contrast to each other, i.e., a natural color crown is very expensive, and an aesthetically less appealing strongly visible dark-colored crown is the most inexpensive possibility for patients. Treatment choices via the choice scenarios in our experiment should therefore clearly show which attributes are most important to patients, and whether they may have to make compromises. 

The SHI standard care treatment might be the most cost-effective alternative to patients as it guarantees a medically necessary treatment in compliance with the principle of economic efficiency, but aesthetically it is an unattractive solution. Due to these circumstances, patient preferences can be determined especially with regard to these two attributes. Furthermore, we see whether cost coverage by SHI is considered sufficient by patients. To our knowledge, there are no studies for the German health care system so far, considering dental crown treatment, using the methodological approach of a discrete-choice-experiment, and focusing patient preferences. To emphasize the special nature and novelty of our study, we have revised the introduction: “We focus on a prosthodontic treatment – the placement of a full dental crown – due to high variability in options and costs to be borne by the patients themselves. In Germany, SHI covers a fixed subsidy of 50% (60% as of 10/2020) for standard treatment of dental crowns. The remaining 50% (40%) have to be paid out of pocket by patients, plus the difference of costs when choosing superior materials. The attributes out-of-pocket payment and aesthetics vary greatly between different dental crown treatments. These assumed (un)desirable attributes for patients behave proportionally against each other, i.e., an aesthetically pleasing dental crown is expensive and vice versa. The SHI alternative may implicate an aesthetically unattractive result to patients (i.e., darker-colored not natural appealing dental crown). To our knowledge, this treatment has not yet been studied for the German health care system using an experimental approach.” (line 76ff.).

2. In Line 89, the authors mention the methods used. I agree that these methods are well understood in the health sciences literature but some discussion on why these are relevant to answer the question of interest would be great.

Thank you for pointing out this. We now included a part on the relevance of DCE and WTP and why it is suitable to answer the research questions: “Discrete-choice experiments (DCE) are an established instrument particularly in health sciences [14] for measuring patient preferences in their choice behavior by estimating benefit assignments, and for calculating willingness-to-pay (WTP) taking out-of-pocket payments into account. Although, medical professionals usually recommend a treatment option, due to restricted coverage in oral health care services, final decisions are largely guided by patient preferences.” (line 94ff.).

3. The experiment provides two unlabeled alternatives and the option of no treatment. However, as the authors note, individuals face multiple choices. Therefore, just providing two choices may not elicit the preferences as there might be behavioral biases in real life. For example, patients might choose a worse alternative in all attributes if they are averse to searching for multiple treatments or if that was marketed better. In this context, it would have been interesting to see if patients respond differently to different size of choice sets.

Thank you for your comment. We completely agree with you that in real life, patients have more than just two or three (in the case of opt-out) treatments to choose from. We set the number of treatment alternatives to n=2, as this is sufficient in an unlabeled design to determine preferences or benefit assignments to attributes. To ensure a balanced distribution of the attributes’ values across all choice sets, we used an experimental design, i.e., its order was output by the SAS software. To ensure evaluability, we might have had to present participants with twice as many choice sets. This would probably have had a negative effect on the willingness to participate but also on the individuals’ concentration when filling in the questionnaires [1]. Nevertheless, we thank you again for this advice. Regarding follow-up studies, we will consider the point raised by you.

4. There could be a selection bias in the response as most individuals included in the study have medium and low household incomes. Hence, the estimated parameters could be biased. Specifically, the authors mention earlier in the text that dental crown treatments might require significant financial resources but since most of the individuals who responded have low household incomes, it raises questions about the external validity of the experiment.

Thank you for the assessment. When planning to send out the questionnaires, we looked at the topic of disposable household incomes in the two focused federal states Berlin and Brandenburg, and its distribution in districts and counties. According to our sources, household incomes are given equally over all Berlin districts in total and the counties of Brandenburg in total, and between the counties. Moreover, number of districts and counties in urban and rural areas in these two states is similar (n=16 and n=14) [2, 3]. In addition, the incomes in Berlin and Brandenburg are close to the national average [4]. The questionnaires were distributed randomly. Thus, we could not influence which income groups were reached and how often. Therefore, we could initially expect to reach all income groups with our questionnaires. To avoid presenting participants extreme values for the attribute out-of-pocket payment, we made graduations here with roughly equal intervals (i.e., 50, 150, and 200, 450, and 600 €, respectively). An acceptable amount of out-of-pocket payment should have been given for each income group.

Unfortunately, we could not influence the household income group of the participants who actually returned the questionnaire and which was included in the analyses. It seems reasonable to assume that people with low incomes were more willing to participate in our survey on dental treatments and its costs in order to express their opinion than people with higher incomes.

However, we have added the point raised by you to the limitations: “When selecting the regional areas to which the questionnaires were sent, we took care to ensure an equal distribution of household incomes across federal states’ districts and counties. Nevertheless, a large proportion of the participants tended to belong to low household incomes groups. This may have biased the results, particularly regarding financial preferences.” (line 494ff.). In addition, we added a note to the conclusion: “Although, the findings should be interpreted with caution due to limitations of choice experiments and the regional restriction to two federal states in Germany, […].” (line 525f.). Furthermore, we have added information on our considerations regarding the household income distribution: “We aimed at an equal distribution of urban and rural population. By prevailing conditions, number of districts and counties in these areas are similar [27]. Household incomes in Berlin and Brandenburg are the same overall and are approximately equally distributed among Brandenburg's counties [28], with incomes in both states close to the national average [29]. Furthermore, we aimed at women and men equally distributed. In compliance with the European General Data Protection Regulation, address data of potential participants, considering the minimum age and an equal gender distribution, were requested according to §34 of the Federal Registration Act from residents' registration offices of the city of Berlin and selected counties in Brandenburg.” (line 157ff.).

5. I am quite confused by the different sample size on the choice model result tables. It would be good to explain why these sample sizes are different.

Thank you for pointing out this. We assume that you are referring to the "no. of observations" that can be found in the results tables. This is not the sample size, but rather the number of data rows that the STATA software included in the analyses. The number of participants whose data records were included in each choice analysis is always the same at n=380. Depending on the question (and thus the model), different numbers of data rows were identified as missings and were not included in the analyses. Therefore, the "no. of observations" varies between the tables. In the case of the ASCL models, the difference in the "no. of observations" to the other tables is particularly large. The reason for this is that only certain choice sets were included in the analysis (i.e., only with SHI standard treatment attributes). Thus, the number of evaluated data rows and thus "no. of observation" has been greatly reduced.

In the results section of the manuscript, we have added a text passage for explanation accordingly: “The "no. of observations" in the (appendix) tables do not refer to the population sample size, but to the dataset rows included in the analyses, and therefore vary between the models. Rows with missings, and of non-relevant choice sets (e.g., SHI levels in ASCL) were excluded.” (line 339ff.).

6. The lack of statistical significance on the results for anterior teeth maybe due to a low sample size. It would be good to send more questionnaires. If not, then it would improve readability to move that table to the appendix.

Thank you for your suggestion. Due to personnel and financial resources, it is unfortunately not possible for us to send additional questionnaires. We fully agree with you that the table section on anterior teeth in Table 4 disturbs the flow of reading. Therefore, we have removed this part of the table from the manuscript and added it as a single table to the appendix (S6 Table).

When calculating the sample size for this discrete-choice experiment, we followed Orme's rule of thumbs [5]. With a number of n=380 participants, we had reached the minimum number of participants, albeit at the lower minimum limit (n=375). Since the rule of thumb was apparently not entirely successful for our study, we therefore added half a sentence to the discussion section (limitations): „In the results of the models, especially ASCL, there is partly no statistical significance, although we have reached the minimum sample size for DCE according to the rule of thumb by Orme [30].” (line 499f.).

7. It is great that the authors discuss the external validity of their experiment, but it would have been interesting to get their take on why that might be a problem in the specific case of dental crowns.

Thank you for the assessment. The external validity was tested by adding a choice set to the questionnaire in which two alternatives with only best vs. worst attribute levels were given (so-called "clear question"). The assumption was that only participants who understood the questionnaire in its content would naturally choose the alternative with the best levels combination. In the case of dental crowns, this best (vs. worst) alternative is particularly easy to determine, since it can be defined as follows: best aesthetics (natural color vs. strongly visible), lowest risk (no risk vs. risk of allergic or local toxic reaction), longest durability (25 years vs. 5 years), and lowest cost (50 € vs. 600 € out-of-pocket payment). In terms of price-performance-ratio, this alternative is assumed to be the best (worst) choice.

We have added a sentence to the methods section explaining what is meant by the “clear question”: “For the "clear question", the two-alternative-choice sets included only the best vs. worst attribute levels (e.g., 25 vs. 5 years durability). It can be assumed that only participants who understood the questionnaire’s content correctly answered this question appropriately.” (line 138ff.).

8. It would be interesting to see how the choice behavior is dependent on the accessibility to dental clinics by the respondent at an administrative district level as that data

Thank you for the assessment. By depicting the "urban" and "rural" area, we have tried to include the participants’ region of residence component in the analyses (e.g., results of regression analyses, line 412f.). Unfortunately, we do not have data on the locations of medical institutions (dental practices, etc.) that provide dental crown treatments. An analysis in this respect would presumably involve data research on locations of medical institutions and would be time-consuming, but definitely is an interesting point to be considered in further studies. 

III Other Minor Comments

1. I found some typos in the text that I mention below.

Thank you very much for the comments on typos in our manuscript! We have revised the typos in the manuscript.

a. Line 54- “methods” instead of “methodes”

b. Line 73 – “decision making” instead of “decisions making”

c. Line 81 – “addressing” instead of “adressing”

d. Line 119 – “scenarios” instead of “szenarios”

e. Line 155 - “backgrounds” instead of “backrounds”

f. Line 246 – “Akaike” instead of “Akaik”

g. Line 303 – “Coefficient” instead of “coefficiant” (Recurring Mistake)

h. Line 383 – “Booklet” instead of “booklet booklet”

2. I also found that there were some formatting errors with the quotes in some places. Rather than having ‘ “text” ‘, I found some places with ‘ ,,text” ‘.

Thank you very much for your very helpful assessment! We have made the corrections of the quotation marks in the manuscript.

References

1. Bech M, Kjaer T, Lauridsen J. Does the number of choice sets matter? Results from a web survey applying a discrete choice experiment. Health Econ. 2011;20:273–86. doi:10.1002/hec.1587.

2. Statistics Berlin Brandenburg. Regional statistics. National accounts - Berlin and Brandenburg 2018. [Amt für Statistik Berlin-Brandenburg. Regionalstatistiken. Volkswirtschaftliche Gesamtrechnungen - Berlin und Brandenburg 2018.] URL: https://www.statistik-berlin-brandenburg.de/regionalstatistiken/r-gesamt_neu.asp?Ptyp=410&Sageb=82000&creg=BBB&anzwer=8 (Accessed: 03/15/2020).

3. Statistics Berlin Brandenburg. Regional statistics. Population - Berlin and Brandenburg 2018. [Amt für Statistik Berlin-Brandenburg. Regionalstatistiken. Bevölkerung - Berlin und Brandenburg 2018.] URL: https://www.statistik-berlin-brandenburg.de/regionalstatistiken/r-gesamt_neu.asp?Ptyp=410&Sageb=12015&creg=BBB&anzwer=6 (Accessed: 03/15/2020).

4. Destatis - Federal Statistical Office. National Accounts of the federal states (redistribution calculation) - Disposable income of private households: federal states, years. 2017. [Destatis - Statistisches Bundesamt. VGR der Länder (Umverteilungsrechnung) - Verfügbares Einkommen der privaten Haushalte: Bundesländer, Jahre. 2017.] URL: https://www-genesis.destatis.de/genesis/online?operation=abruftabelleBearbeiten&levelindex=1&levelid=1662814398480&auswahloperation=abruftabelleAuspraegungAuswaehlen&auswahlverzeichnis=ordnungsstruktur&auswahlziel=werteabruf&code=82411-0001&auswahltext=&wertauswahl=225&wertauswahl=1111&werteabruf=Werteabruf#abreadcrumb. (Accessed: 09/11/2022).

5. Orme BK. Getting started with conjoint analysis: strategies for product design and pricing research: 2nd ed.: Madison, WI: Research Publishers, 2010.

---

## [Editor Report · Decision Letter 1]

2 Jan 2023

Patients’ preferences in dental care: a discrete-choice experiment and an analysis of willingness-to-pay

PONE-D-22-18370R1

Dear Dr. Felgner,

We’re pleased to inform you that your manuscript has been judged scientifically suitable for publication and will be formally accepted for publication once it meets all outstanding technical requirements.

Kind regards,

Ted Loch-Temzelides

Academic Editor

PLOS ONE

Additional Editor Comments (optional):

The authors did a good a job in addressing the referee's comments. The small sample size and the resulting statistical significance issues remain, but this is not something that could be addressed in this study and it is best left to future research. Some minor expositional suggestions follow:

1. Although you are studying several different regressions, perhaps it is best to label the corresponding sections (e.g., line 402) as "regression analysis" instead of using the plural form.

2. Line 61. Consider using: "quality of life, and reduced productivity"

3. Line 82. Consider replacing "behave" with "move"

4. Line 86. Consider eliminating "Therefore"

5. Lines 110-114 read somewhat repetitive. Please consider rewriting.

6. Line 252. Consider replacing: "unrealistic" with "strong"

7. Line 428 Consider using: "... two teeth areas, PT and AT. The focus of..."

8. Line 456. Consider replacing "Nevertheless, high co-payments would be paid by them, especially for AT" with "Nevertheless these would require high co-payments, especially for AT"

9. Line 471. Consider using: "...income, financial wealth..."

---

## [Editor Report · Acceptance letter]

6 Jan 2023

PONE-D-22-18370R1 

Patients’ preferences in dental care: a discrete-choice experiment and an analysis of willingness-to-pay 

Dear Dr. Felgner:

I'm pleased to inform you that your manuscript has been deemed suitable for publication in PLOS ONE. Congratulations! Your manuscript is now with our production department. 

Kind regards, 

on behalf of

Dr. Ted Loch-Temzelides 

Academic Editor

PLOS ONE